# RNA-guided assembly of Rev-RRE nuclear export complexes

Yun Bai[1], Akshay Tambe[1], Kaihong Zhou[1,2], Jennifer A Doudna[1,2,3,4]*

[1]Department of Molecular and Cell Biology, University of California, Berkeley, Berkeley, United States; [2]Howard Hughes Medical Institute, University of California, Berkeley, Berkeley, United States; [3]Department of Chemistry, University of California, Berkeley, Berkeley, United States; [4]Physical Biosciences Division, Lawrence Berkeley National Laboratory, Berkeley, United States

**Abstract** HIV replication requires nuclear export of unspliced and singly spliced viral transcripts. Although a unique RNA structure has been proposed for the Rev-response element (RRE) responsible for viral mRNA export, how it recruits multiple HIV Rev proteins to form an export complex has been unclear. We show here that initial binding of Rev to the RRE triggers RNA tertiary structural changes, enabling further Rev binding and the rapid formation of a viral export complex. Analysis of the Rev-RRE assembly pathway using SHAPE-Seq and small-angle X-ray scattering (SAXS) reveals two major steps of Rev-RRE complex formation, beginning with rapid Rev binding to a pre-organized region presenting multiple Rev binding sites. This step induces long-range remodeling of the RNA to expose a cryptic Rev binding site, enabling rapid assembly of additional Rev proteins into the RNA export complex. This kinetic pathway may help maintain the balance between viral replication and maturation.

*For correspondence: doudna@berkeley.edu

Competing interests: The authors declare that no competing interests exist.

## Introduction

Intron-containing RNA transcripts are usually retained in the nucleus until they are either spliced or degraded (*Luo and Reed, 1999*; *Bousquet-Antonelli et al., 2000*; *Zhou et al., 2000*). In contrast, during the late stage of HIV infection, unspliced viral transcripts must be exported to the cytoplasm either to be translated into structural proteins or to serve as genomic RNA packaged into new virions (*Cullen, 2003*). To ensure such nuclear export, HIV transcripts assemble into ribonucleoprotein (RNP) particles in which multiple copies of the viral Rev protein, a translation product of fully spliced HIV transcripts, bind to the Rev-response element (RRE) located within a viral RNA intron (*Malim et al., 1989*; *Pollard and Malim, 1998*; *Cullen, 2003*). Upon forming a nuclear–export complex with Crm1 and RanGTP, the Rev-RRE RNP translocates across the nuclear pore into the cytoplasm (*Fornerod et al., 1997*; *Pollard and Malim, 1998*; *Cullen, 2003*; *Yedavalli et al., 2004*).

Formation of the Rev-RRE RNP involves binding of multiple copies of Rev to viral transcripts. Rev can bind RNA as well as self-associate to form dimers and higher-order oligomers. It also contains a nuclear export sequence (NES) that can be recognized by Crm1. Both the RNA binding and dimerization/multimerization properties of Rev play key roles in Rev-RRE complex assembly (*Mann et al., 1994*; *Pond et al., 2009*; *Daugherty et al., 2010a, 2010b*). Initial association of Rev with Stem IIB of the RRE (*Heaphy et al., 1990, 1991*; *Cook et al., 1991*; *Huang et al., 1991*; *Kjems et al., 1991*; *Malim and Cullen, 1991*) leads to additional Rev protein association with a secondary binding site near Stem IA (*Daugherty et al., 2008*) (*Figure 1A*). A total of at least six copies of the Rev protein are thought to be necessary to form a functional RNP, although the exact number of Rev proteins in the mature complex remains a matter of debate (*Mann et al., 1994*; *Daugherty et al., 2008*; *Robertson-Anderson et al., 2011*).

**eLife digest** HIV is a virus that causes the immune system of an infected person to gradually fail, which can eventually result in AIDS. The virus consists of an RNA molecule—which encodes its genetic information—surrounded by coats of proteins. Once HIV enters a host cell, its RNA genome is converted into a DNA molecule, which travels to the nucleus and becomes part of the host's genome. The integrated viral genome can remain dormant for an extended period before the virus starts to replicate.

HIV replication begins with the production of RNA copies of the viral genome. For certain types of viral RNA molecules to be translated and packaged into new virus particles they need to be exported from the nucleus as part of the 'nuclear–export complex'. This is made up of: a HIV RNA molecule, a HIV protein called Rev, and two host proteins.

Formation of the nuclear–export complex begins with multiple copies of the Rev protein attaching to specific stretches of the viral RNA, but how the Rev proteins assemble on the RNA molecule was previously unclear. Bai et al. have now used both structural and biochemical techniques to dissect the individual steps in this process. First, Rev proteins rapidly bind to a pre-formed region of the RNA molecule where multiple binding sites are compactly organized. This causes the overall shape of the RNA to change, and exposes a previously hidden extra binding site for Rev proteins. More Rev proteins then quickly bind to the newly exposed site, before finally the two host proteins bind and the whole complex is exported from the nucleus.

Bai et al. propose that checkpoints during this two-step assembly process are required to ensure that Rev proteins specifically bind to viral RNAs, and that such checkpoints may be important for controlling viral replication. The findings of Bai et al. may, in future, help to develop new drugs that treat HIV infection by blocking the export of the virus from the nucleus and thus inhibiting HIV replication.

Several lines of evidence suggest that Rev assembles into the Rev-RRE complex in a sequential manner (*Lam et al., 1998*; *Van Ryk and Venkatesan, 1999*; *Daugherty et al., 2008, 2010a*; *Pond et al., 2009*; *Robertson-Anderson et al., 2011*). Some evidence suggests that Rev is recruited to the RRE one molecule at a time (*Pond et al., 2009*), while other data are consistent with recruitment in the form of a dimeric complex (*Daugherty et al., 2008, 2010b*). Studies of the Rev-RRE assembly pathway have focused on the state of the RNA and protein components either alone or in the fully assembled complex (*Cook et al., 1991*; *Mann et al., 1994*; *Battiste et al., 1996*; *Charpentier et al., 1997*; *Watts et al., 2009*; *Daugherty et al., 2010b*), or on the dynamics of Rev protein within the complex (*Van Ryk and Venkatesan, 1999*; *Pond et al., 2009*). In the absence of information about structural changes in the RRE that might occur during association with Rev, contradicting Rev-RRE assembly models were proposed (*Daugherty et al., 2010b*; *Fang et al., 2013*). To address this problem, we interrogated the folding pathway of the RRE RNA during the course of Rev-RRE complex formation, both as a function of time and Rev association state. We find that three regions within the RRE, bridged by previously unidentified RNA tertiary interactions, exhibit sequential structural changes during RNA-protein assembly. Comparison of the kinetic and thermodynamic pathways of Rev-RRE complex formation suggests a two-step assembly mechanism beginning with Rev binding to a pre-organized RRE structure critical for enhanced binding rate. This interaction triggers rearrangement of long-range RNA contacts to facilitate higher-order Rev-RRE assembly. This complex promotes a switch to late HIV protein expression and packaging.

## Results and discussion

### SHAPE-based secondary structure model and tertiary interactions of the RRE RNA

To determine the structural features of an extended RRE transcript including sequences beyond the primary and secondary Rev binding sites, we performed selective 2'-hydroxyl acylation analyzed by primer extension (SHAPE) analysis (*Merino et al., 2005*) of the 354-nt RRE RNA of HIV-1 isolate ARV-2/SF2 (*Figure 1A*). This RNA segment was chosen because it forms an independently folded region flanked by unstructured sequences within the HIV genome (*Watts et al., 2009*). We determined its SHAPE

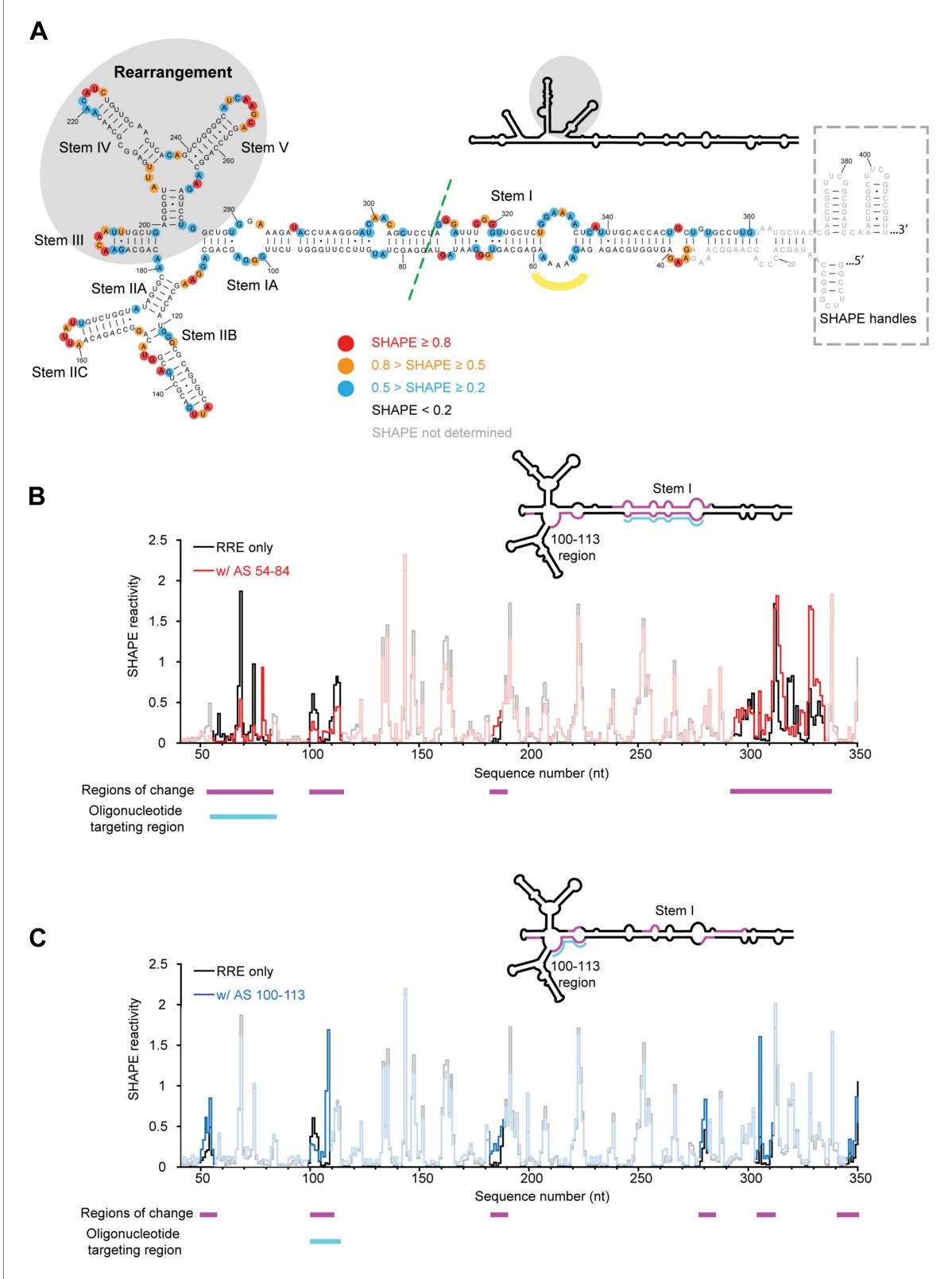

**Figure 1**. HIV RRE RNA adopts a pre-organized compact fold. (**A**) SHAPE-based secondary structure of the RRE RNA. Red, orange, and blue dots highlight nucleotides with high, medium, and low SHAPE reactivity, respectively. Nucleotides with no SHAPE reactivity are in black. Nucleotides with SHAPE reactivity unidentified are in gray. The box shows the SHAPE handles in the RNA construct. The region of secondary structure rearrangement in

*Figure 1. Continued on next page*

*Figure 1. Continued*

our prediction is highlighted by gray shadow. The commonly used secondary structure is shown in the upper left with the rearranged region shadowed. The green line shows the ends of the 233-nt RRE construct (*Fang et al., 2013*). The yellow curve indicates the position of nucleotides 54-58. (**B**) SHAPE profiles of the RRE RNA alone (black) and the RNA-oligo complex (red) with AS 54-84. The anti-sense oligo region is shown in cyan, and the regions of SHAPE reactivity change are shown in magenta. Those regions are labeled both under the data chart and on the secondary structure model. (**C**) SHAPE profiles of the RRE RNA alone (black) and the RNA-oligo complex (blue) with AS 100-113. The anti-sense oligo region is shown in cyan, and the regions of SHAPE reactivity change are shown in magenta. Those regions are labeled both under the data chart and on the secondary structure model.

The following figure supplements are available for figure 1:

**Figure supplement 1**. Designed oligonucleotides can invade and hybridize to the RRE RNA at specific sites.

**Figure supplement 2**. SHAPE changes induced by oligonucleotides interactions.

profile and used this information for secondary structure prediction on the RNAstructure Web Server (*Low and Weeks, 2010*; *Reuter and Mathews, 2010*). The resulting SHAPE-based secondary structure model is largely consistent with current models for the RRE, including that obtained from structural probing of the entire HIV-1 genome (*Mann et al., 1994*; *Charpentier et al., 1997*; *Daugherty et al., 2008*, *2010a*; *Pond et al., 2009*; *Watts et al., 2009*; *Fang et al., 2013*). We note that the SHAPE data predict a rearrangement in the Stem III/IV and Stem V regions relative to previous models, yielding a modified secondary structure more similar to that of the RRE in SIV (*Pollom et al., 2013*), which is used for mapping additional structural features and protein interactions detected in this study (*Figure 1A*).

Surprisingly, although predicted to be in a loop region, nucleotides 54-58 exhibit either low or no SHAPE reactivity, indicating that they could be constrained by tertiary contacts. To test whether this loop forms long-range interactions with another part of the RRE, we designed a 31-nt oligonucleotide complementary to nucleotides 54-84 (AS 54-84), which can efficiently invade and hybridize with the pre-folded RRE RNA to disrupt its local structure, and determined the SHAPE profile for the resulting RNA-oligo complex (*Sztuba-Solinska and Le Grice, 2012*). The length of this oligonucleotide was determined experimentally to be the minimum required for stable association with the RNA, presumably due to competing stability of the RRE secondary structure in this region (*Figure 1—figure supplement 1A*). In addition to the Stem I region (around nucleotides 50-80 and 300-340), which was directly affected by the oligonucleotide binding, another segment covering nucleotides 100-113 showed significant changes in SHAPE reactivity, even though it is distant from the target region based solely on predicted secondary structure (*Figure 1B*, *Figure 1—figure supplement 2A*). Results from toe-printing assays showed that AS 54-84 did not bind to a secondary site on the RRE (*Figure 1—figure supplement 1B*), indicating the SHAPE change could be a result of long-range crosstalk. To confirm this, a 14-nt oligonucleotide complementary to nucleotides 100-113 (AS 100-113) was used in a reciprocal experiment to perturb its potential long-range contacts. A shorter oligonucleotide was used here because this segment is predicted to be more accessible for oligonucleotide hybridization based on thermodynamic predictions. Toe-printing assays again showed no secondary oligonucleotide binding on the RRE (*Figure 1—figure supplement 1A,B*). This resulting RNA-oligo complex showed substantially altered SHAPE reactivity in several patches along the Stem I region, including patches spanning around nucleotides 50, 310, and 340 (*Figure 1C*, *Figure 1—figure supplement 2B*). Therefore, disruption of the tertiary structure in the Stem I region specifically affects the 100-113 segment and the converse is also observed. These results suggest the existence of a long-range structural contact between Stem I and nucleotides 100-113. No sequence complementarity exists between these two regions, indicating that any tertiary interaction may be more complex than direct base-pairing.

## The RRE RNA adopts a compact structure

Based on observations above, we hypothesized that tertiary interactions bridging Stem I and the 100-113 region could make the overall fold of the 354-nt RRE relatively compact. This is consistent with results of a recent SAXS study, which revealed that a 233-nt RRE construct adopts an A shape with a maximum diameter ($D_{max}$) of ~195 Å (*Fang et al., 2013*). Similarly, we found that the 354-nt RRE construct used here has a $D_{max}$ of 198 Å as detected by SAXS (*Figure 2A–D*). This result suggests that the long Stem I in the RRE folds back towards the core of the multi-way junction instead of forming an extended tail (*Figure 2E*).

If the relatively compact conformation of the RRE results from the tertiary interactions identified earlier, disruption of those interactions is predicted to result in a more extended conformation. To test

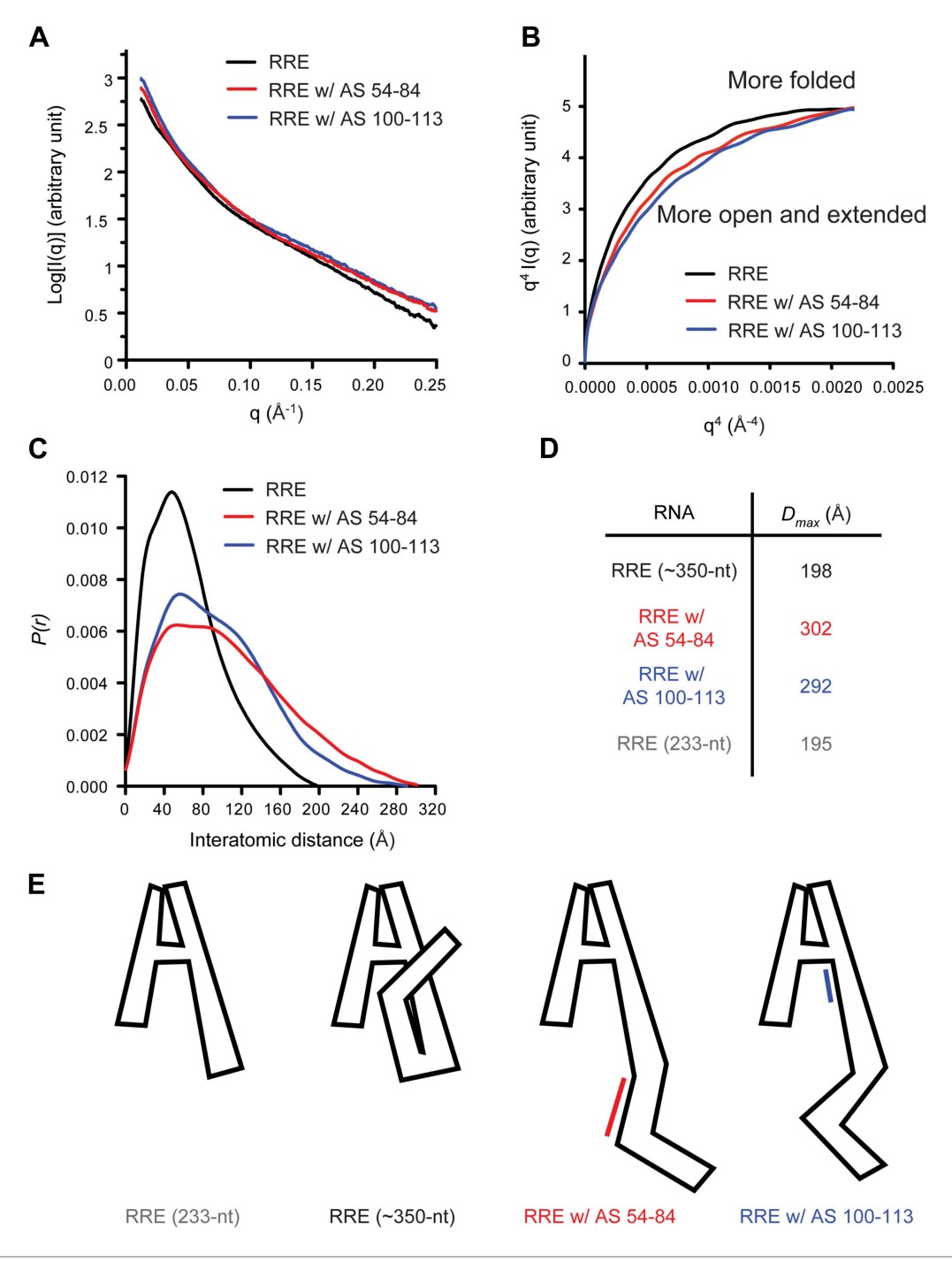

**Figure 2**. Model of the compact RRE RNA conformation. (**A**) RRE-oligo complexes show scattering patterns different from that of the RRE RNA alone. (**B**) Porod-Debye plot of RRE RNA and RNA-oligo complexes indicates the native RNA is more folded and the RNA-oligo complexes are more open and extended. (**C**) Distance distribution function ($P(r)$) of RRE RNA and RNA-oligo complexes. (**D**) Comparison of particle maximum diameter ($D_{max}$) of RRE RNA and RNA-oligo complex. Data for 233-nt RNA is as published (***Fang et al., 2013***). (**E**) Model for the compact fold of the RRE mediated by tertiary interactions between the 100-113 region and Stem I.

The following figure supplement is available for figure 2:

**Figure supplement 1**. Guinier plots of the SAXS data.

this, we used the antisense oligonucleotides described above (AS 54-84 or AS 100-113) to interfere with long-range interactions within the RRE and performed SAXS measurements on the resulting complexes. As expected, the RNA-oligo complexes showed markedly altered scattering patterns (*Figure 2A*, *Figure 2—figure supplement 1*; *Supplementary file 1*) and became more open and extended compared to the native RRE based on the Porod-Debye plot (*Rambo and Tainer, 2011*; *Figure 2B*). The $D_{max}$ values for the resulting complexes also increased dramatically to 302 Å and 292 Å, respectively. It has been reported that a 30-nt single-stranded DNA oligonucleotide shows a $D_{max}$ value of ~90 Å under similar $Mg^{2+}$ concentration (*Meisburger et al., 2013*), indicating that even if free oligonucleotide remained in the system after extensive washes, it would not bias the distance distribution functions (*P(r)*) to a longer range. Therefore, the large variations in the *P(r)* indicate a substantial conformational change in the RRE (*Figure 2C,D*). Taken together, these results demonstrate that the Stem I and 100-113 regions are linked through long-range interactions, which enables the RRE to fold into a compact structure (*Figure 2E*). We speculate that the sharp bend in the RNA (*Figure 2E*) results from a series of bends that occur throughout the noncanonical parts of Stem I.

## Thermodynamic pathway of Rev-RRE RNP assembly

To determine how the pre-formed RRE structure affects RNP assembly, we used SHAPE to locate all Rev binding-induced structural changes on the RRE RNA and associate them with individual Rev binding events. Experiments were conducted on equilibrated complexes formed using different Rev:RRE ratios (*Figure 3A*). In parallel, the percentage of different sub-complexes within each binding reaction was quantified by electrophoretic mobility shift assays (EMSAs) using aliquots from the same samples (*Figure 3B*, *Figure 3—figure supplement 1A*).

From the EMSA data, we detected the formation of different Rev bound sub-complexes as a function of Rev:RRE stoichiometry (*Figure 3B*), and used that information to deduce distinct SHAPE modification signatures that represent progressive Rev binding states (*Figure 3—figure supplement 1B*; 'Materials and methods'). We then performed *k*-means clustering (*Eisen et al., 1998*) on nucleotides with significant SHAPE reactivity changes (ΔSHAPE > 0.15) ('Materials and methods') to group them together according to SHAPE signatures. All SHAPE signatures fall into seven clusters, each containing a group of nucleotides sharing a common pattern of SHAPE reactivity changes as a function of Rev concentration (*Figure 3C*). Based on the SHAPE signatures and the Rev bound sub-complexes they represent, nucleotides within the seven clusters are associated with RNA structural changes triggered by Rev binding at low, intermediate and high Rev:RRE stoichiometries and those regions are named Region1, Region2 and Region3, respectively (*Figure 3D*).

Region1 covers the primary, high-affinity Rev binding site reported previously (*Heaphy et al., 1990*, *1991*; *Cook et al., 1991*; *Huang et al., 1991*; *Kjems et al., 1991*; *Malim and Cullen, 1991*). It also includes the three-way junction of Stems IIA, IIB and IIC. According to earlier reports, Region1 is likely recognized by a Rev dimer (*Zemmel et al., 1996*), with one molecule binding in the widened RNA major groove at the primary binding site (*Battiste et al., 1996*) and the other binding at the three-way junction (*Zemmel et al., 1996*; *Van Ryk and Venkatesan, 1999*; *Daugherty et al., 2008*) (Jayaraman et al., unpublished manuscript). Region2 covers the previously identified secondary Rev binding site (*Daugherty et al., 2008*). By comparing the footprint of this region with that of Region1 together with a previous truncation study (*Van Ryk and Venkatesan, 1999*), we infer that Region2 could accommodate at least a Rev dimer. Region3, a previously undefined Rev binding site, is located in the center of Stem I (*Figure 3D*). It comprises an array of purine-rich bulges, which resemble the preferred RNA site for Rev binding (*Heaphy et al., 1991*; *Tan et al., 1993*; *Battiste et al., 1996*). In fact, Region3 assignment is consistent with previous reports showing that positions on Stem I can be protected by Rev oligomers, and truncations in this area affect oligomeric Rev binding (*Mann et al., 1994*; *Robertson-Anderson et al., 2011*). Interestingly, Region3 also overlaps with the area on Stem I that could form long-range interactions with the nucleotides 100-113 within Region2 (*Figure 3D*, *Figure 1B,C*). Although Region2 and Region3 are distant from each other based on secondary structure, our observations indicate that tertiary interactions could bring them into close proximity, which might facilitate Rev binding at those regions while forming a continuous oligomer at the same time (*Figure 2E*).

## Kinetic assembly pathway of the Rev-RRE complex

To follow the dynamics of Rev-RRE RNP formation, we performed time-resolved SHAPE to examine how the RRE RNA changes during the course of RNP assembly (*Figure 4A*). To obtain snapshots of the

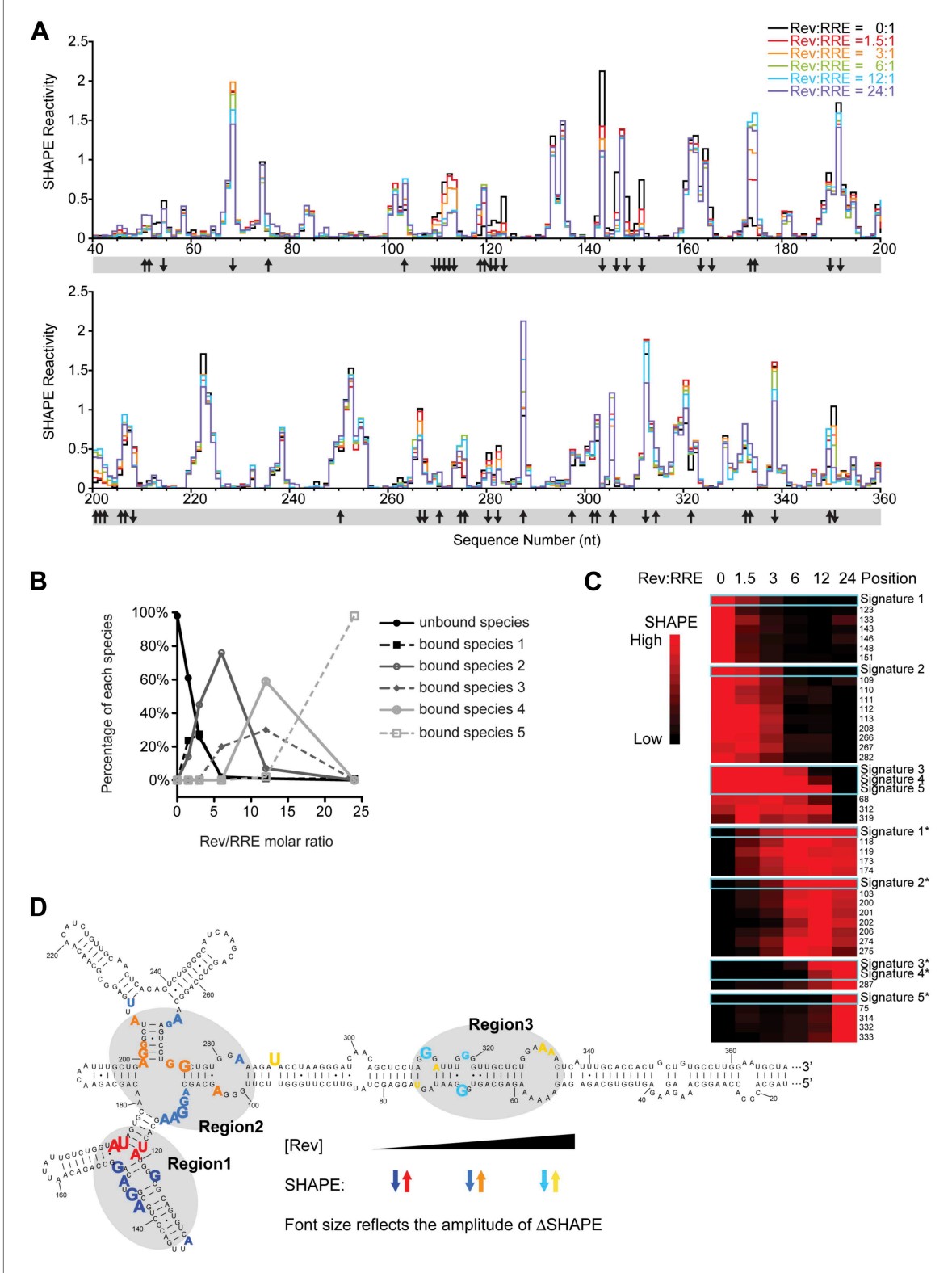

Figure 3. Thermodynamic studies on the Rev-RRE assembly pathway. (A) SHAPE profiles from samples at different Rev:RRE ratio. The positions showing increase/decrease of SHAPE reactivity are indicated with arrows at the bottom of the plots. (B) Trend of emergence of each Rev-RRE sub-complex at increasing ratios of Rev. (C) k-means clustering result of nucleotides following distinct SHAPE signatures with the SHAPE signatures shown in cyan boxes. 5 of the 10 SHAPE signatures (Signatures*) represent increased SHAPE reactivity as a function of Rev concentration, while the other five (Signatures)
*Figure 3. Continued on next page*

*Figure 3. Continued*

reflect decreased SHAPE reactivity. They fall into seven clusters because some signatures are not distinct enough from others. Red indicates higher SHAPE reactivity while black indicates lower SHAPE reactivity. (**D**) Nucleotide positions affected by Rev binding. In red, orange and yellow shows positions with increased SHAPE reactivity upon Rev binding. In dark, medium and light blue shows positions with decreased SHAPE reactivity upon Rev binding. Changes emerging at lower Rev:RRE stoichiometry are shown in darker colors, while changes emerging at higher Rev:RRE stoichiometry are shown in lighter colors. Font size reflects the amplitude of SHAPE change. Region1, Region2 and Region3 are highlighted by gray shadows.

The following figure supplement is available for figure 3:

**Figure supplement 1**. SHAPE signatures generated by EMSA.

RNA on a timescale of seconds, a fast-reacting SHAPE reagent, benzoyl cyanide, was used in these experiments (*Mortimer and Weeks, 2008*, *2009*). Based on a previous single-molecule study of the Rev-RRE system, the timescale for each Rev binding step is ~2–5 s (*Pond et al., 2009*). This indicates that under our experimental conditions, intermediate stages of RNP assembly should be detectable. To increase the precision of the measurements, we adapted the SHAPE-Seq method (*Lucks et al., 2011*) to determine SHAPE reactivities and rates of SHAPE-monitored structural changes at different positions on the RRE. The accuracy of this method depends on the number of reads used for calculating each SHAPE profile. In this study, the amount of data for one replicate at each time point range from 256,424 to 1,145,824 reads. With this amount of data, the reactivity values (theta, the modification probability [*Lucks et al., 2011*]) are highly reproducible with Pearson correlation coefficients ranging from 0.967 to 0.974 between any two of the three replicates (*Figure 4—figure supplement 1*).

Using this method of analysis, distinct dynamic features were observed at different regions on the RRE during RNP assembly as a function of time (*Figure 4A,B*). At most nucleotide positions, the SHAPE-reactivity change patterns follow single exponential decay. However, more complex SHAPE kinetic patterns are observed for several other nucleotides, most of which show an increase of SHAPE reactivity at earlier time points followed by a gradual decrease (*Figure 4C*). Next, we calculated the rate for all positions showing a significant SHAPE change ($\Delta$SHAPE > 0.15) by fitting the time-resolved data using exponential decay/association kinetics. Only the earlier time points for nucleotides with complex kinetic behavior were used for fitting in our initial comparison (*Supplementary file 2*). During Rev-RRE assembly, structural changes of the RRE originate from the primary Rev binding site and subsequently propagate along the RNA. The pattern is similar to that observed based on the thermodynamic experiments described above (*Figure 3D*, *Figure 4B*). Even though the accuracy of the fastest SHAPE reactivity change rates is limited by the time resolution of the experiment, the overall trend of sequential SHAPE changes is evident (*Figure 4B*).

Upon encountering Rev, Region1 exhibits SHAPE reactivity changes consistent with previous structural data for complexes of Rev and RNA fragments covering this region (*Battiste et al., 1996*) (Jayaraman et al., unpublished manuscript). At this time resolution, the two Rev binding events that occur in this region, one at Stem IIB and one at the three-way junction at Stem IIA, IIB and IIC, are indistinguishable. And the majority of SHAPE reactivity changes at Region1 occur within 1 s (*Figure 4D*; *Supplementary file 2*). The overall rates of SHAPE reactivity changes at Region2 are only marginally slower than those observed at Region1, and the half-life for the majority of changes in Region1 and Region2 are clustered together below 1.5 s (*Figure 4D*; *Supplementary file 2*). Since the RRE RNA at this local region is pre-organized into a compact fold (*Fang et al., 2013*; *Figure 2E*), its conformation can facilitate Rev multimerization from Region1 to Region2 with little RNA rearrangement necessary, leading to rapid and highly cooperative binding of multiple Rev proteins. Based on previous reports, binding of the third Rev molecule (the second Rev dimer) and beyond requires higher oligomerization capability of the protein (*Daugherty et al., 2008*). Therefore, Rev self-association should also be a fast process that can be completed within the same time period.

In contrast, Region3 shows by far the slowest rate of folding, with the initial stage of most SHAPE changes in this region showing half-lives of 2–12 s (*Figure 4D*; *Supplementary file 2*). All nucleotides with complex kinetic behaviors are located in this region, and the second phase of their SHAPE reactivity change continues beyond 50 s at many positions (*Figure 4A*, *Figure 4—figure supplement 2B*). Combined with data revealing the tertiary fold of the RRE, for the nucleotides with complex kinetic behaviors, the increase in SHAPE reactivities at earlier time points is consistent with rearrangement of the RRE

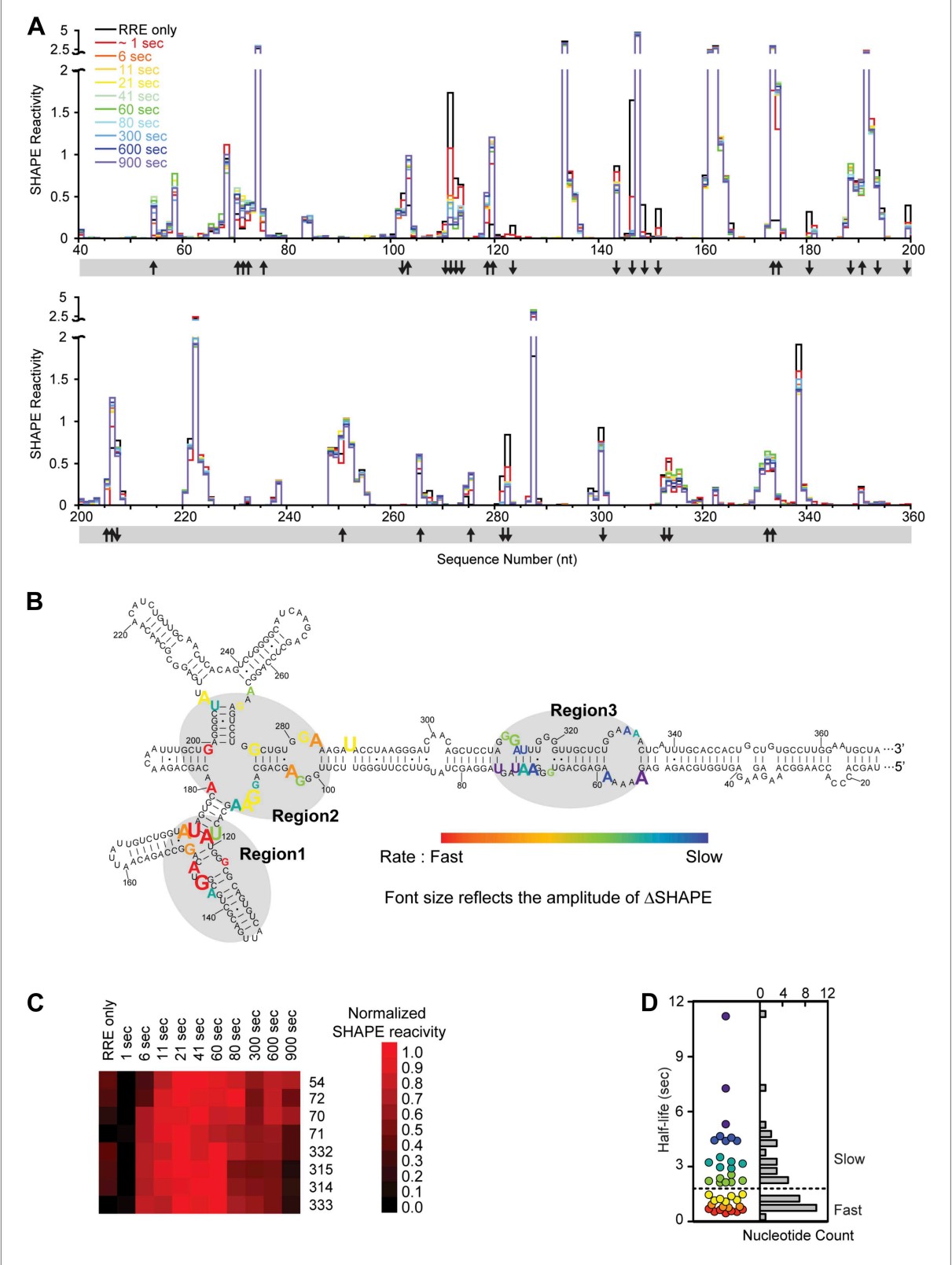

**Figure 4**. Dynamic assembly of the Rev-RRE RNP. (**A**) SHAPE profiles for second-resolution snapshots of the RRE at different time points over the course of Rev-RRE assembly. The positions showing increase or decrease of SHAPE reactivity are indicated with arrows at the bottom of the plots. For ease of comparison, the theta values are converted to SHAPE reactivity values by renormalization following the 2%/8% rule for each SHAPE profile, with the top

*Figure 4. Continued on next page*

*Figure 4. Continued*

2% of theta values being excluded and the next 8% of theta values being averaged to get the normalization factor, against which all the theta values are normalized (*Low and Weeks, 2010*; *Lucks et al., 2011*). (**B**) Nucleotide positions affected by Rev binding. Different nucleotides are labeled across the spectrum based on the rate of SHAPE change, with the fastest rate in red and the slowest in purple. Font size reflects the amplitude of SHAPE change. Region1, Region2 and Region3 are highlighted by gray shadows. (**C**) Heat map showing the SHAPE pattern as a function of time for the nucleotides with complex SHAPE-changing features. For each nucleotide in this panel, the SHAPE values are normalized to 0–1 in order to emphasize the trend of SHAPE change. (**D**) Scatterplot (left) and histogram (right) showing the distribution of the SHAPE change half-life for nucleotides showing significant SHAPE change. Colors used in the scatter plot correspond to those used in panel **B**. Both plots show a fast-SHAPE-changing group and a slow-SHAPE-changing group.

The following figure supplements are available for figure 4:

**Figure supplement 1**. Quality and reproducibility test for the SHAPE-Seq data.

**Figure supplement 2**. Rev-RRE assembly process exhibits two-step features.

---

tertiary structure at Region3. This could occur after Rev disrupts the long-range contact within the RRE by binding to Region2; the subsequent decrease in SHAPE reactivity could represent additional Rev binding in this region (*Figure 4C*). The slower rates of structural change in Region3 could be due to RNA conformational changes, or adjustment of Rev conformation as influenced by surrounding RNA and proteins to allow additional Rev oligomerization, or a combination of both effects. Nonetheless, characteristics exhibited by this stage of assembly best resemble an induced-fit model of RNA-protein recognition.

Due to the slow assembly of the higher-order Rev-RRE complex, the full RNP formation could take minutes (*Figure 4—figure supplement 2B*). The latter phase of SHAPE reactivity changes at those positions is much noisier (*Figure 4—figure supplement 2B*). This could reflect a combination of 'conformational selection' and 'induced-fit' events, leading to a series of specific and non-specific contacts that facilitate finding the optimal binding configuration, similar to observations for other RNP assembly pathways (*Bokinsky et al., 2006*; *Boehr et al., 2009*; *Rau et al., 2012*; *Kim et al., 2014*).

## Tertiary folding of the RRE facilitates accelerated RNP assembly

Previously, it has been reported that in the absence of Region3, a truncated ~240-nt RRE can mediate HIV RNA nuclear export but with lower efficiency (*Malim et al., 1989*; *Huang et al., 1991*), indicating a facilitating role of the sequences outside of the ~240-nt RRE. These observations are consistent with our results showing that Region3 on the extended Stem I (*Figure 1A*) is a preferred Rev binding location in the context of the full-length RRE. Moreover, sequences on the extended Stem I also mediate tertiary interactions within the RRE. We then asked besides providing additional Rev binding site, whether the extended Stem I region contribute to Rev-RRE complex assembly, especially at the early stage, due to the tertiary interactions it mediates.

SHAPE snapshots were taken at 0, 1, 6, 11, and 900 s of the complex assembly on three mutants of the extended Stem I. One of them is a 242-nt truncation mutant and the other two are site mutants in Region3 alone (Region3 mut1) and Region3/Stem I (mut2), which reduce the flexibility of Stem I by introducing additional base-pairing (*Figure 5A*). SHAPE reactivity changes at key nucleotides in Region1 and Region2 were compared between different constructs and the 354-nt RRE. For the ease of comparison, the SHAPE reactivity values were normalized so that zero represents the SHAPE state for unbound RRE at 0 s and 1 represents the SHAPE state of the RRE in the fully assembled complex at 900 s. At 6 and 11 s, all RNA constructs behave similarly, and the normalized SHAPE changes indicate that Rev binding at Region1 and Region2 are mostly completed at those time points. In contrast, at 1 s the 354-nt RRE shows overall higher normalized SHAPE changes than all three mutants, indicating that Rev binding rate is enhanced in the context of the full-length RRE with intact Stem I region (*Figure 5B,C*). This enhancement is observed at early Rev binding sites, underlining the importance of the RNA tertiary folding. These results reveal that the extended Stem I of the RRE, which covers a cryptic Rev binding site, can enhance the rate of the Rev-RRE complex assembly, likely by pre-organizing the RRE into a favorable conformation for Rev binding. Assembly of this RNP governs the balance for nuclear export of different types of HIV transcripts, which is essential for promoting HIV infectious cycle. Offset in the assembly rate of the Rev-RRE complex could break the optimal balance of cytoplasmic HIV transcripts, which not only affect the distribution of HIV transcripts but could also make a greater difference in the encapsidation of genomic RNA into infectious viral particles (*Brandt et al., 2007*).

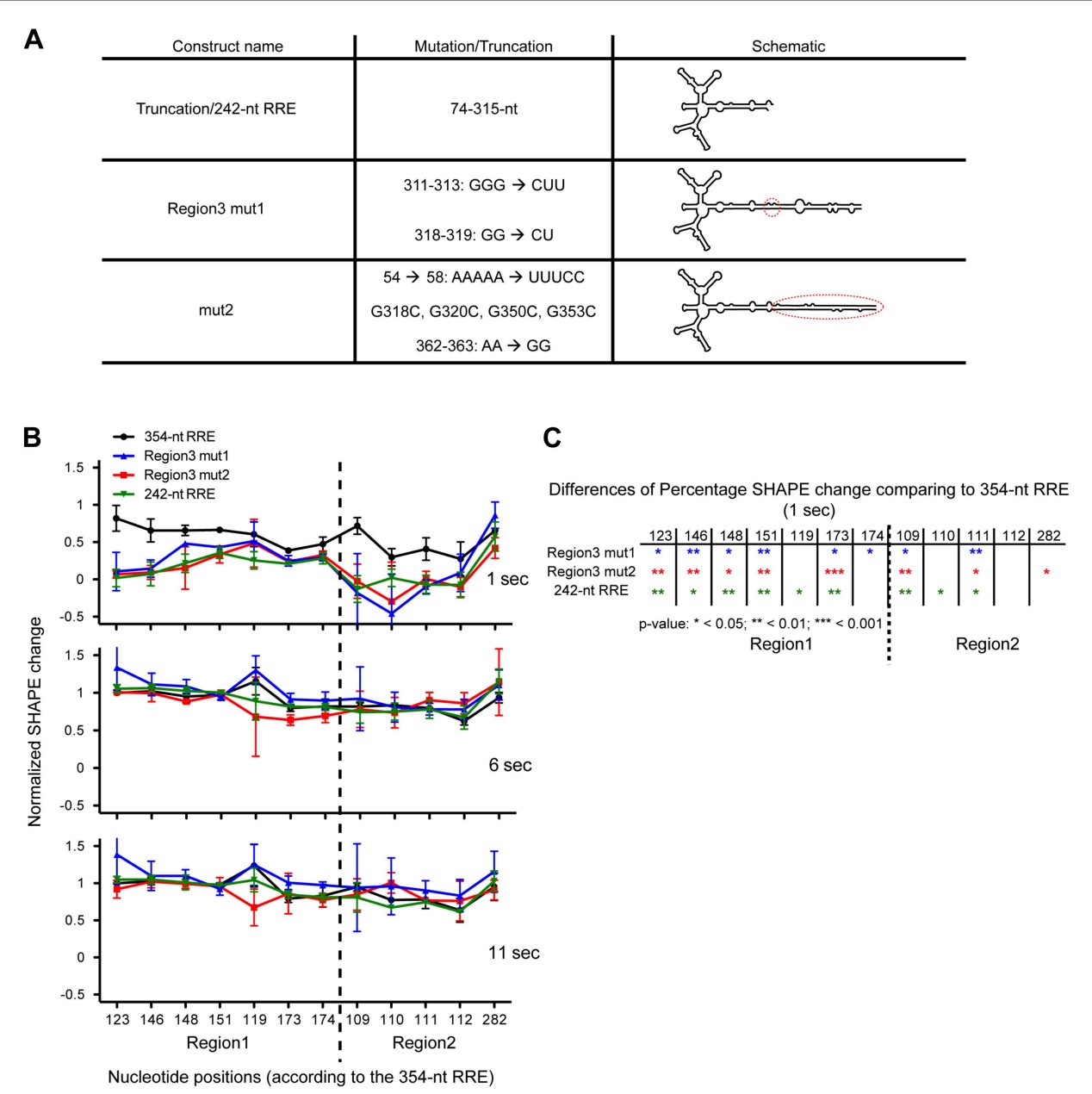

**Figure 5.** Functional importance of the Region3. (**A**) Table showing the mutations used in this figure. (**B**) Normalized SHAPE change at 1, 6 and 11 s after Rev binding. The ~354-nt RRE is shown in black, Region3 mut1 is shown in blue, mut2 is shown in red and the 242-nt RRE is shown in green. (**C**) Table indicating the significance of the differences in normalized SHAPE changes between different constructs at 1 s. All mutants are compared to the 354-nt WT RRE. DOI: 10.7554/eLife.03656.013

Together, these results fill in a long missing piece of this RNP assembly puzzle: how the Rev-response element structure responses to the addition of Rev in a dynamic manner. Our data suggest a concerted Rev-RRE complex assembly mechanism and indicate how specificity can be achieved here with limited components. We propose that Rev-RRE assembly features two distinct stages (*Figure 4D*, *Figure 4—figure supplement 2A*). The first stage largely utilizes pre-organized RNA structure for protein recruitment, while the second stage involves more global RNA conformational changes and induced-fit RNA-protein recognition (*Figure 4D*, *Figure 6*). The similarity in the thermodynamic and kinetic RNP assembly pathways indicates that this process is hierarchical, with the RRE RNA driving RNP assembly by organizing sequential Rev binding (*Figure 6*). The highest affinity Rev-binding site, Stem

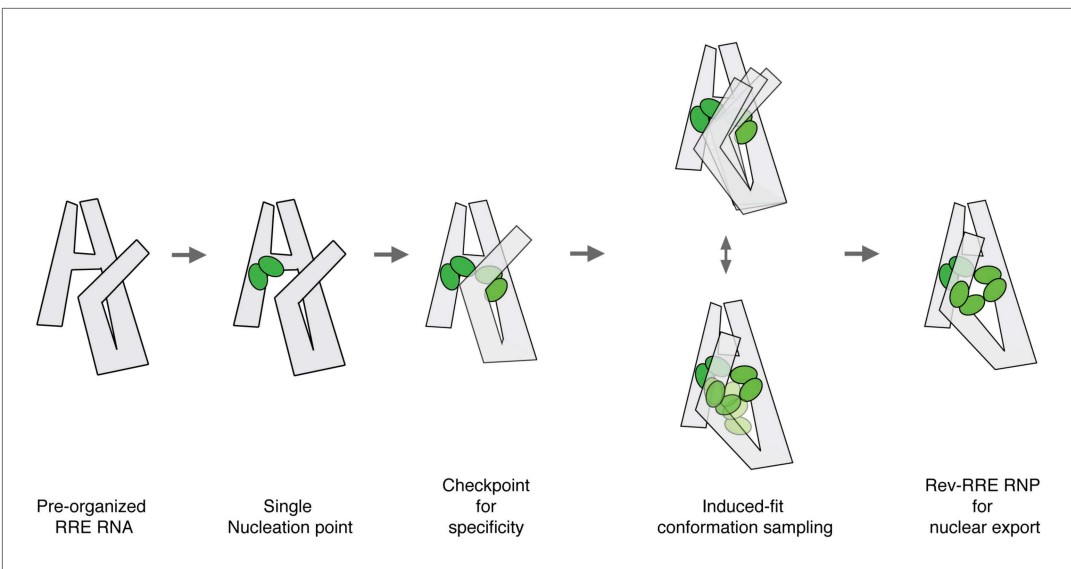

**Figure 6**. Model for pre-organized RRE RNA guides sequential binding of Rev to form the Rev-RRE RNP. RRE RNA forms a compact fold in the absence of Rev. Rev assembly on the RRE starts from a single nuclection point. Region1 and Region2 binding are coupled and the four-Rev complex state can serve as a checkpoint to ensure specificity. Region2 Rev binding leads to conformational change of the RRE to allow additional Rev binding through induced-fit. Both Rev and RRE could sample a number of interaction conformations at the same time until an optimal binding state is reached. The high-oligomer complex is then ready for Crm1 binding and nuclear export.

IIB, ensures that complex assembly nucleates from a single origin on the RNA. Binding of up to four Rev proteins at Region1 and Region2 is tightly coupled to obtain a stable intermediate state (*Zemmel et al., 1996*; *Van Ryk and Venkatesan, 1999*; *Daugherty et al., 2008*; *Pond et al., 2009*). This step also releases a cryptic Rev binding site at Region3, which can make further contacts with additional Rev molecules. The tertiary folding of the RRE facilitates the efficiency of Rev-RRE complex formation. In addition, the hierarchical nature of the assembly could ensure its selectivity and accuracy of the final complex with limited number of components. These features together provide a fine control over cytoplasmic distribution of various HIV transcripts as well as viral packaging. Intriguingly, these steps in Rev-RRE assembly resemble those that occur during bacterial ribosome assembly in which stably formed rRNA structures recruit initial ribosomal protein binding partners. The resulting local structures trigger long-range RNA conformational rearrangements that enable binding of additional ribosomal proteins (*Adilakshmi et al., 2008*; *Shajani et al., 2011*; *Kim et al., 2014*). Such similarities suggest that common mechanisms for RNP assembly could be shared among diverse biological processes. Properties of the HIV RNP assembly pathway elucidated here present opportunities for antiviral strategies that could block the nuclear export step of HIV replication by targeting important intermediates of the Rev-RRE complex (*Fenster et al., 1994*; *Chapman et al., 2002*; *Shuck-Lee et al., 2008*; *Ward et al., 2009*).

## Materials and methods

### Sample preparation

The RRE RNA construct used here contains the 354-nt full-length RRE from isolate ARV-2/SF2 with SHAPE handles at both ends as previously reported (*Berry et al., 2011*). RNA samples from all constructs were prepared by in vitro transcription using T7 polymerase, with either linearized plasmid or PCR product DNA as templates. For SHAPE analysis RNA samples were column-purified using RNA Clean & Concentrator-25 (Zymo Research, Irvine, CA, USA), and for SAXS experiments RNA samples were gel-purified and washed multiple times through filtration. Purified RNA samples were annealed in a buffer containing 50 mM HEPES-KOH pH 7.5, 200 mM KOAc, and 3 mM $MgCl_2$ by heating at 75°C for 2 min and snap cooling on ice. For SHAPE-Seq experiments, barcodes on the RRE molecules were introduced by PCR and placed within the 3′ SHAPE handle as previously described (*Lucks et al., 2011*;

*Mortimer et al., 2012*). Both the His-GB1-Rev fusion protein construct and its purification procedure were reported previously (*Daugherty et al., 2008*, *2010a*).

## Binding and SHAPE reactions

SHAPE probing was performed as previously reported (*Bai et al., 2013*) with 40 mM of benzoyl cyanide (BzCN) (Sigma-Aldrich, St. Louis, MO, USA) used as the 2′ hydroxyl-selective electrophile. To make RRE-oligonucleotide complexes, RRE RNA was first annealed at 0.1 mg/ml followed by a 5 min incubation at room temperature. A large excess of oligonucleotides complementary to either nucleotides 54-84 (AS 54-84) or nucleotides 100-113 (AS 100-113) was added to pre-folded RRE and the mixture was incubated for 20 min at room temperature to allow the oligonucleotides to form a complex with the RNA. For SAXS measurements, the RRE-oligonucleotide complexes were washed extensively by ultrafiltration. To make Rev-RRE complex at different stoichiometries, GB1-Rev fusion protein was first diluted to various concentrations (0.25–4 mg/ml) in Rev buffer containing 40 mM Tris pH 8.0, 200 mM NaCl, 100 mM $Na_2SO_4$, 400 mM $(NH_4)_2SO_4$, 2 mM β-ME, and 10% glycerol. 30 µl of annealed RRE RNA at 0.1 mg/ml was then mixed with 3 µl of either Rev protein solution or Rev buffer alone. The resulting mixtures were incubated for 20 min at room temperature. 10 µl of each resulting solution was analyzed by EMSAs, while the proceeding SHAPE protocol was used on the remaining sample. SHAPE-Seq experiments were performed as described (*Mortimer et al., 2012*) with slight modification of the primer design, as listed below. 10 parts of annealed RRE RNA at 0.1 mg/ml was mixed with one part of 4 mg/ml Rev protein and SHAPE reaction was performed at different time points. For the ~1 s time point, the SHAPE reagent was introduced at the same time with the Rev protein by centrifugation and the ~1 s time is estimated based on the half-life of BzCN (*Mortimer and Weeks, 2008*). The RRE RNA samples used in this assay are barcoded within the SHAPE handle as described (*Lucks et al., 2011*) and additional barcodes were introduced during the PCR amplification step of library preparation using the NEBNext Multiplex Oligos for Illumina kit (NEB, Ipswich, MA, USA) to increase the multiplex capacity.

> RT_index1:
> AATGATACGGCGACCACCGAGATCTACACTCTTTCCCTACACGACGCTCTTCCGATCTNNNNNNN
> RRRYGAACCGGACCGAAGCCCG
> RT_index2:
> AATGATACGGCGACCACCGAGATCTACACTCTTTCCCTACACGACGCTCTTCCGATCTNNNNNNN
> YYYRGAACCGGACCGAAGCCCG
> A_adapter_b_short:
> 5′ Phos-ATGCNNNNNNNNNAGATCGGAAGAGCACACGTCTGAACTCCAGTCAC-C3
> Paired_end_reverse:
> CAAGCAGAAGACGGCATACGAGATGTGACTGGAGTTCAGACGTGTGCTCTTCCGATCT

## SHAPE analysis

For capillary electrophoresis based experiments, raw traces from fragment analysis were analyzed using ShapeFinder (*Vasa et al., 2008*). For the RRE-oligonucleotide complexes, samples treated with only DMSO but not SHAPE reagent were also analyzed as toe-printing assays to determine the binding sites of the oligonucleotides. For sequencing based experiments, raw data was processed using FASTX-Toolkit (http://hannonlab.cshl.edu/fastx_toolkit/). A quality filter was applied so that only reads with a minimum of 95% bases having a quality score of over 30 were retained for data analysis. Paired-end reads were aligned to the RRE RNA using Bowtie 0.12.7 (*Langmead et al., 2009*) and SHAPE reactivity was determined using Spats (*Aviran et al., 2011b*; *Lucks et al., 2011*; *Aviran et al., 2011a*). Reads with identical sequences were not collapsed during our data process due to file format compatibility consideration. However, primer ID (*Jabara et al., 2011*) was included in the primers to control for PCR amplification bias. Only <3% of the reads are from PCR duplication and most of them are aligned to adapter dimers. Therefore, there is no significant PCR bias in our experiment.

## SHAPE-based secondary structure prediction

SHAPE-based secondary structure of the RRE RNA was calculated using RNAstructure Web Servers (*Reuter and Mathews, 2010*) with the SHAPE reactivity file obtained for the free RRE RNA (*Low and Weeks, 2010*). The SHAPE Intercept and SHAPE Slope used for this prediction were −0.6 and 1.8, respectively.

### Generating SHAPE signatures

SHAPE signatures for each Rev binding event were generated based on EMSA results. First, the trend of emergence for each of the RRE or Rev-RRE complex was derived directly from the quantification of the EMSA data. With the assumption that a SHAPE change associated with an earlier Rev binding event remains in all subsequent complexes, we next calculated a series of SHAPE signatures, which reflect the emergence of RNP with one Rev and above, two Rev and above, etc. This series of curves were used to represent positions showing increased SHAPE reactivity upon Rev binding. Subsequently, another series of curves were generated reflecting the disappearance of species with less than one Rev, less than two Rev, etc. These series of curves were used to represent positions showing decreased SHAPE reactivity.

### *k*-means clustering

*k*-means clustering of SHAPE profile at different nucleotide positions was performed using *Cluster* 3.0 (*Eisen et al., 1998*; *de Hoon et al., 2004*) as previously reported (*Grohman et al., 2013*) with slight modification to also include that SHAPE signatures as guidance points. Results were visualized using Java TreeView (*Saldanha, 2004*). Here ΔSHAPE > 0.15 is considered as significant SHAPE change. However, cutoffs of ΔSHAPE > 0.2 and ΔSHAPE > 0.1 gave qualitative similar result after data analysis.

### SHAPE-reactivity changing rate determination

SHAPE-reactivity changing rates for nucleotides showing significant SHAPE change (ΔSHAPE > 0.15) from time-resolved SHAPE-Seq experiments were determined by fitting the data to either one-phase association or one-phase decay function with automatic outlier elimination using GraphPad Prism 5.0b for Mac OS X.

### Small-angle X-ray scattering

SAXS data were collected at the Advanced Light Source (Lawrence Berkeley National Laboratory) beamline 12.3.1 (*Hura et al., 2009*; *Classen et al., 2013*). Two-dimensional scattering curves obtained at different exposure time were merged and processed using PRIMUS (*Konarev et al., 2003*) and distance distribution functions *P(r)*, radius of gyration (Rg) and Porod volume were generated using GNOM (*Svergun, 1992*). Porod-Debye plot was calculated based on raw data using GraphPad Prism 5.0b for Mac OS X.

## Acknowledgements

We want to thank members of the Frankel laboratory, JD Fernandes for technical assistance, DS Booth for providing Rev protein samples, plasmids and purification protocols and B Jayaraman and AD Frankel for sharing results before publication; M Chung at QB3 Vincent J Coates Genomics Sequencing Laboratory for performing the Illumina sequencing; SA Mortimer, SN Floor and ASY Lee for help in data analysis; MA Kidwell for help with figure making; AD Frankel, J Fernandes, DS Booth, B Jayaraman and members of the Doudna lab for helpful discussions and critical reading of the manuscript. The SAXS data was collected at SIBYLS Beamline 12.3.1 at the Advanced Light Source of Lawrence Berkeley National Laboratory. This project was supported in part by the HARC Center National Institutes of Health Grant P50GM82250 (to JAD), and by the Howard Hughes Medical Institute. JAD is a Howard Hughes Medical Institute investigator.

## Additional information

### Funding

| Funder | Grant reference number | Author |
|---|---|---|
| National Institute of General Medical Sciences | The HARC Center, P50GM82250 | Jennifer A Doudna |
| Howard Hughes Medical Institute | | Jennifer A Doudna |

The funders had no role in study design, data collection and interpretation, or the decision to submit the work for publication.

## Author contributions

YB, Conception and design, Acquisition of data, Analysis and interpretation of data, Drafting or revising the article; AT, KZ, Acquisition of data, Contributed unpublished essential data or reagents; JAD, Conception and design, Drafting or revising the article

## Additional files

### Supplementary files

• Supplementary file 1. Comparison of RRE alone and RRE in complex with antisense oligonucleotides.

• Supplementary file 2. SHAPE-changing rate demonstrating half-life at different nucleotides. For positions with complex SHAPE reactivity changes, only the six earlier data points were used for the fitting. Data was fitted to one-phase decay/association with automatic outlier elimination using GraphPad Prism 5.0b for Mac OS X.

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
