## [Decision Letter]

[Editors’ note: although it is not typical of the review process at *eLife*, in this case the editors decided to include the reviews in their entirety for the authors’ consideration as they prepared their revised submission.]

Thank you for sending your work entitled “RNA-guided assembly of Rev-RRE nuclear export complexes” for consideration at *eLife.* Your article has been favorably evaluated by James Manley (Senior editor) and 4 reviewers, one of whom is a member of our Board of Reviewing Editors.

The Reviewing editor and the other reviewers discussed their comments before we reached this decision, and the Reviewing editor has assembled the following comments to help you prepare a revised submission.

We apologize for the time taken to handle your manuscript but one of the reviewers was delayed in submitting. In any case, we now have comments from four expert reviewers. As you will see, all were quite positive about the work but three reviewers, especially reviewers 3 and 4, raise some points that should be addressed via revision. Please address these as thoroughly as possible, without necessarily performing additional experimental work.

Reviewer #1:

This manuscript elucidates the kinetic pathway whereby the Rev/RRE nuclear export complex is assembled. Using a variety of techniques including SHAPE and SAXs it was determined that initial Rev binding to a structured element induces a conformational shift in the RRE which reveals a heretofore unreported “cryptic” Rev biding site. The data are presented clearly and they are convincing. Their study provides important insight into HIV biology in particular and RNA protein interactions in general. I have no major (or minor) concerns and publication is recommended as is.

Reviewer #2:

This is a very well written and executed study that provides fundamental new insights into the mechanisms that underlie the assembly of Rev-RRE nuclear export complexes through a combination of equilibrium and time-resolved SHAPE, SAXS, and other biochemical experiments. It identifies nucleating points, dynamic RNA tertiary interactions and RNA structure remodeling that together promote the specificity and cooperativity of assembly. I have only minor suggestions for consideration.

Minor comments:

1) Formation of the 3y interactions seems to require a very sharp 180 degree turn in a helix-junction as shown in Figure 2 – the authors may want to speculate how this might be accomplished – is there a particular junction (bulge or loop ( that can accommodate such a sharp bend or might this be the result of many smaller bends about a large number of closely spaced junctions? Could this be the result of direct interactions with Rev?

2) The authors may want to speculate on the reverse process of Rev-RRE disassembly and whether the mechanisms seen here for assembly, if reversed, could confer additional functionality in the context of disassembly.

Reviewer #3:

This is a very well written paper describing various sets of elegant experiments on the formation of Rev-RRE complexes. My concerns are the following. Can the authors exclude the presence of RNA dimers? Have they done native gels? There are neighboring stretches of Us that could pair to the protected set of As (especially in a dimer). It would be also valuable to see the Shape analysis and SAXS profiles for the mutant like mut2 but limited to the 54-58 region (with data shown as in Figure 1). What are the values for the Rg Guinier and Porod volume from the SAXS data? These values may be relevant for the state of the RNA in solution. Finally, on a personal note, I think chemical probing with DMS (revealing the accessibility of the Watson-Crick positions) would add very convincing evidence to the Shape data that reflect only the backbone.

Reviewer #4:

The authors investigate long range remodeling of the HIV viral transcript (Rev response element) located within viral RNA intron upon binding with viral Rev protein. Analysis of the Rev-RRE assembly pathway using SHAPE-Seq and small-angle X-ray scattering (SAXS) reveals two major steps of Rev-RRE complex formation. The study shows how Rev-RRE formation begins with rapid Rev binding to a reorganized region presenting multiple Rev binding sites.

The authors present a unique protocol that allows evaluation and visualization of conformational changes in RRE under the Rev-protein or antisense - RRE-oligos binding. This SHAPE-SAXS combo approach is a novel and useful approach for determining how functional conformational changes triggers RNA tertiary structural changes, enabling further Rev binding and the rapid formation of a viral export complex. The authors also present a model of dynamic RRE changes under REV oligomerisation on RRE for a set of time-resolved SHAPE data sets.

Overall, this is an excellent study and the conclusions are based upon stated quality results. Although the SAXS structure of truncated RRE was determined by Xianyang Fang et al. (2013, Cell), this study brings novel and valuable information about dynamic behavior of RRE-REV formation. Besides the novelty in the biology, the presented SHAPE-SAXS approach is novel and may be applied for other nucleoprotein complexes. The authors should address the following points:

1) The solution Scattering analysis could be improved by added missing SAXS quality checks:

a) A Mw estimation from the SAXS will help to eliminate possibility of the formation of larger RRE assemblies (Nature 496, 477-481, 2013).

b) Guinier plots will define the aggregation state of the sample.

2) Xianyang [21] presented SAXS envelopes of RRE-223nt that have been further interpreted with a simplified RNA-duplex model. It will be useful to see if the SAXS data presented here for RRE∼350nt reassembles this shape. Even with the larger flexibility in stem loop 1, it may be possible to visualize the RRE∼350 and conformational disorder in the stem loop1 in presence of antisense oligos 54-84, 100-113.

3) “SAXS data were collected at the Advanced Light Source (Lawrence Berkeley National Laboratory) beamline 12.3.1.” A relevant reference here would provide information for readers who otherwise will not know the experimental details, e.g. Journal of Applied Crystallography 46, 1-13, 2013 or Nature Methods 6, 606-612, 2009 depending upon how the experiments were done.

4) Why are there no SAXS experiments on Rev-RRE nucleoprotein complexes where REV-RRE was successfully studied in detail with SHAPE.

---

## [Author Response]

Reviewer #2:

*This is a very well written and executed study that provides fundamental new insights into the mechanisms that underlie the assembly of Rev-RRE nuclear export complexes through a combination of equilibrium and time-resolved SHAPE, SAXS, and other biochemical experiments. It identifies nucleating points, dynamic RNA tertiary interactions and RNA structure remodeling that together promote the specificity and cooperativity of assembly. I have only minor suggestions for consideration*.

Minor comments:

*1) Formation of the 3y interactions seems to require a very sharp 180 degree turn in a helix-junction as shown in*
Figure 2
*– the authors may want to speculate how this might be accomplished – is there a particular junction (bulge or loop ( that can accommodate such a sharp bend or might this be the result of many smaller bends about a large number of closely spaced junctions? Could this be the result of direct interactions with Rev?*

We suspect that the sharp turn in Stem I is a combined result of the series of internal loops and bulges present in the RRE. Our mutagenesis studies (Figure 5) show that replacing these internal loops and bulges with a perfectly based-paired duplex results in a reduced rate of Rev-RRE RNP assembly. This effect is similar to that observed in the 242-nt truncated RRE. Direct interactions with Rev are not required, since the tertiary contacts associated with the bent RNA are present in the absence of Rev. To clarify this in the text, we have added the following sentence: “We speculate that the sharp bend in the RNA (Figure 2) results from a series of bends that occur throughout the noncanonical parts of Stem I.”

*2)The authors may want to speculate on the reverse process of Rev-RRE disassembly and whether the mechanisms seen here for assembly, if reversed, could confer additional functionality in the context of disassembly*.

With our current data, it is really difficult to speculate how Rev-RRE disassembly works, especially since there could be other cellular factors involved. This interesting question is a subject of ongoing studies.

Reviewer #3:

*This is a very well written paper describing various sets of elegant experiments on the formation of Rev-RRE complexes. My concerns are the following. Can the authors exclude the presence of RNA dimers? Have they done native gels? There are neighboring stretches of Us that could pair to the protected set of As (especially in a dimer)*.

Under the experimental conditions used, migration of the RRE RNA in native gels is consistent with that of a monomer. We also do not detect concentration-dependent changes in native gel mobility of the RNA, as would be expected if the RNA were dimeric. In addition, the SHAPE data do not show protection of the consecutive Us as would be expected if they were involved in dimer formation.

It would be also valuable to see the Shape analysis and SAXS profiles for the mutant like mut2 but limited to the 54-58 region (with data shown as in Figure 1). What are the values for the Rg Guinier and Porod volume from the SAXS data? These values may be relevant for the state of the RNA in solution. Finally, on a personal note, I think chemical probing with DMS (revealing the accessibility of the Watson-Crick positions) would add very convincing evidence to the Shape data that reflect only the backbone.

In mut2, nucleotides 54-58 were mutated to base pair with the other side of that internal loop. This region shows low SHAPE reactivity in the wildtype RRE and there were no pronounced differences in this region in mut2. We agree that SAXS measurements of RRE mutants for comparison to the wildtype RRE would be interesting, but these experiments are beyond the scope of the present study.

We now include the Guinier plot, the Rg from GNOM anaylsis and the Porod volume as [Supplementary-material SD1-data]. The shape and large size of the molecule as well as the SAXS beamline configuration used for data collection limited the range of low q data, which affects the accuracy of Rg estimation from Guinier analysis. We note that the observed linear Guinier region indicates that the samples were not aggregated. In this case, the Rg from GNOM analysis should better represent the sample properties, and we compare these values in the new supplementary table. The limited low q data is the reason we did not carry out ab initio modeling using the current data set.

Reviewer #4:

*The authors investigate long range remodeling of the HIV viral transcript (Rev response element) located within viral RNA intron upon binding with viral Rev protein. Analysis of the Rev-RRE assembly pathway using SHAPE-Seq and small-angle X-ray scattering (SAXS) reveals two major steps of Rev-RRE complex formation. The study shows how Rev-RRE formation begins with rapid Rev binding to a reorganized region presenting multiple Rev binding sites*.

*The authors present a unique protocol that allows evaluation and visualization of conformational changes in RRE under the Rev-protein or antisense - RRE-oligos binding. This SHAPE-SAXS combo approach is a novel and useful approach for determining how functional conformational changes triggers RNA tertiary structural changes, enabling further Rev binding and the rapid formation of a viral export complex. The authors also present a model of dynamic RRE changes under REV oligomerisation on RRE for a set of time-resolved SHAPE data sets*.

*Overall, this is an excellent study and the conclusions are based upon stated quality results. Although the SAXS structure of truncated RRE was determined by Xianyang Fang et al. (2013, Cell), this study brings novel and valuable information about dynamic behavior of RRE-REV formation. Besides the novelty* in *the biology, the presented SHAPE-SAXS approach is novel and may be applied for other nucleoprotein complexes. The authors should address the following points:*

1) The solution Scattering analysis could be improved by added missing SAXS quality checks:

*a) A Mw estimation from the SAXS will help to eliminate possibility of the formation of larger RRE assemblies (Nature 496, 477-481, 2013)*.

*b) Guinier plots will define the aggregation state of the sample*.

Guinier plots, real space Rg and Porod volume are now included as [Supplementary-material SD1-data]. The Porod volume is consistent with the size of monomers. Mw estimation according to the reference above was not performed because the Vc plot (from Scatter) shows discontinuity at low q value, likely because we do not have sufficient data at low q value for a molecule of this size.

*2) Xianyang*
[21]
*presented SAXS envelopes of RRE-223nt that have been further interpreted with a simplified RNA-duplex model. It will be useful to see if the SAXS data presented here for RRE∼350nt reassembles this shape. Even with the larger flexibility in stem loop 1, it may be possible to visualize the RRE∼350 and conformational disorder in the stem loop1 in presence of antisense oligos 54-84, 100-113*.

We agree it would be informative to compare the two models. Modeling the full-length RRE as well as mapping different domains within it using various truncations and tagging strategies is an interesting direction for future experiments but is not within the scope of this study.

*3) “SAXS data were collected at the Advanced Light Source (Lawrence Berkeley NationalLaboratory) beamline 12.3.1.” A relevant reference here would provide information for readers who otherwise will not know the experimental details, e.g. Journal of Applied Crystallography 46, 1-13, 2013 or Nature Methods 6, 606-612, 2009 depending upon how the experiments were done*.

The suggested references are included in the revised manuscript.

*4) Why are there no SAXS experiments on Rev-RRE nucleoprotein complexes where REV-RRE was successfully studied in detail with SHAPE*.

We did not perform those experiments due to challenges in preparing homogeneous samples suitable for SAXS analysis. Since SHAPE measurements reflect the average state of the population of molecules, non-specific interactions of Rev and the RNA only contribute to background signal. Future experiments, beyond the scope of the present study, are aimed at generating samples suitable for structural analysis, including by SAXS.